# Reservoirs of Biodiversity: Gardens and Parks in Portugal Show High Diversity of Ivy Species

**DOI:** 10.3390/plants14162486

**Published:** 2025-08-11

**Authors:** Pedro Talhinhas, João da Cunha Ferreira, Ana Paula Ramos, Ana Luísa Soares, Dalila Espírito-Santo

**Affiliations:** 1LEAF—Linking Landscape, Environment, Agriculture and Food Research Centre, Associate Laboratory TERRA, Instituto Superior de Agronomia, Universidade de Lisboa, Tapada da Ajuda, 1349-017 Lisboa, Portugal; pramos@isa.ulisboa.pt (A.P.R.); dalilaesanto@isa.ulisboa.pt (D.E.-S.); 2Instituto Superior de Agronomia, Universidade de Lisboa, Tapada da Ajuda, 1349-017 Lisboa, Portugal; 3LPVVA—Laboratório de Patologia Vegetal Veríssimo de Almeida, Instituto Superior de Agronomia, Universidade de Lisboa, Tapada da Ajuda, 1349-017 Lisboa, Portugal; 4CEABN—Centre for Applied Ecology “Professor Baeta Neves”, InBIO, Instituto Superior de Agronomia, Universidade de Lisboa, Tapada da Ajuda, 1349-017 Lisboa, Portugal; alsoares@isa.ulisboa.pt

**Keywords:** *Hedera*, garden, biodiversity, Portugal

## Abstract

Urban parks and gardens are important in multiple ways, but they are often mostly made of exotic species with little biodiversity value. Surveys conducted in public and private gardens and parks across diverse Portuguese cities reveal a surprisingly high level of ivy species diversity, even when there is no apparent ornamental value in growing multiple species. The analysis of 499 samples from Mainland Portugal, Azores, and Madeira shows that in Madeira, the endemic *H. maderensis* co-occurs with exotic species; in Mainland Portugal, *H. hibernica* is more common in the centre and north (49% of samples) and *H. iberica* in the south (50% of samples), following their distribution in nature, but co-occur with exotic species (mostly *H. helix*, *H. algeriensis*, *H. maroccana*, and *H. canariensis*). Often, different species are cultivated side by side in the same garden, thus depicting these gardens as hidden reservoirs of biodiversity and, simultaneously, as potential sources for biological invasion.

## 1. Introduction

Urban parks and gardens are increasingly recognised as vital components of urban ecosystems, serving not only recreational and aesthetic functions but also acting as reservoirs of biodiversity. As urbanisation intensifies, natural habitats are fragmented or lost, placing pressure on native species and ecological processes. However, green spaces within cities—such as parks, gardens, and remnant natural areas—offer refuge for a wide range of flora and fauna, contributing to ecological resilience and the conservation of biodiversity in anthropogenic landscapes. Recent studies have demonstrated that urban green spaces can support surprisingly high levels of species richness, particularly when designed with ecological principles in mind, including diverse vegetation structures and habitat connectivity [1]. Moreover, the presence of biodiversity in urban environments has been linked to positive outcomes for human well-being, emphasising the multifaceted value of these spaces [2]. In this context, the role of urban parks and gardens in Portugal merits closer examination, particularly regarding their capacity to host diverse plant species.

Among the ornamental plants with significant ecological and historical value, ivies (*Hedera* spp.) stand out, being widely used in Portuguese historical gardens as well as in different types of public and private gardens. These evergreen climbing plants are appreciated in ornamental horticulture for their ornamental traits, ground and wall-covering abilities, and resilience, being quite tolerant to drought, as well as for providing shelter and food for urban wildlife.

The genus *Hedera*, comprising 13 species, exhibits notable morphological and genetic diversity, with several species and cultivars adapted to different climatic and ecological conditions (e.g., [3]). Phylogenetic studies have revealed that ivies form a polyploid complex with a rich biogeographic history, originating in Eurasia and North Africa and attaining the highest of its diversity in its westernmost native range (more than half of the species are native from this area: *H. azorica*, *H. maderensis*, *H. canariensis*, *H. maroccana*, *H. algeriensis*, *H. iberica*, *H. helix*, and *H. hibernica*), and are now present in gardens worldwide [4]. Nevertheless, ivies have shown invasive behaviour in regions outside their natural range, namely, in North America [5]. Recent studies have demonstrated that *Hedera helix* can negatively impact native understory regeneration and alter soil chemistry in invaded ecosystems [6], while playing equally important roles in its native ecosystems, but with these roles depicted as positive in such cases [7]. In areas where ivies are native, the invasive status of alien *Hedera* spp. becomes difficult to assess because of difficulties in species identification and changes in taxonomy over time [8].

The cultivation of ivies dates to Classical Antiquity, and they were referred to as ornamental in the Iberian Peninsula by Arabs [9], with exotic species of North African origin (*H. algeriensis* and *H. maroccana*) recorded as widely cultivated and even naturalised [10]. Portuguese gardens, particularly in urban and Mediterranean contexts, are characterised by a blend of cultural influences and ecological adaptation, which is expressed in the distinct selection of plant species, often incorporating drought-tolerant and shade-resilient species, including several *Hedera* varieties. However, only a few species were developed into cultivars [11]. Such cultivars are available in the market for cultivation, in some circumstances replacing native species, posing a risk of biological invasion. When possible, the cultivation of native species is, in principle, preferred over that of similar exotic species, a situation that is particularly evident in the case of ivies. The ability of ivies to disperse over long distances and occupy a wide range of environments regarding altitude, sun exposure, soil preferences, and low water requirements for irrigation makes the use of ivies in practical applications of landscape architecture projects beneficial, as their successful growth requires minimal maintenance [11,12].

This study aims to assess ivy species diversity cultivated in Portuguese parks and gardens, with the objective of evaluating the role of these green spaces as reservoirs of biodiversity. By focusing on the genus *Hedera*, we seek to highlight the ecological significance of ornamental plants in urban ecosystems and to underscore the importance of integrating native species into landscape planning. The findings are expected to contribute to a deeper understanding of how urban green infrastructure can support plant diversity, inform sustainable horticultural practices, and reinforce the conservation value of public and private gardens in Portugal.

## 2. Results

In this study, we collected and identified to species level a total of 499 ivy samples from parks and gardens in Portugal. These included six samples from the Azores (Faial and São Miguel islands) and 37 from Madeira, as well as the following number of samples per district in mainland Portugal: Aveiro—7; Beja—6; Braga—7; Bragança—3; Castelo Branco—5; Coimbra—3; Évora—16; Faro—7; Guarda—1; Leiria—21; Lisboa—288; Portalegre—8; Porto—8; Santarém—42; Setúbal—25; Viana do Castelo—1; Vila Real—6; Viseu—2. In total, this comprised 25 samples from the north of the country, 86 from the centre, and 345 from the south (Figure 1).

Most samples were collected in private gardens (167 samples) and in public gardens (142), followed by home backyards (88) and public parks (63). A smaller number of samples was collected from roadside flower beds (11), pots (6), or at the edge of industrial plots (10).

Ivies were most frequently growing as dominant plants (63% of samples), and in only 10% of samples were ivies dominated by other plants. Two-thirds of the sampled ivies were growing vertically only, whereas 14% were growing horizontally only, and the remaining 20% were growing both horizontally and vertically. Vertical growth included walls and fences, as well as trees and artificial vertical structures. A total of 37 samples presented variegated leaves.

With 180 samples, *Hedera iberica* was the most common species (36% of samples), followed by four species representing 13–15% of the occurrences: *H. helix* (77 samples); *H. hibernica* (72); *H. algeriensis* (69); and *H. maroccana* (67). The Madeira and Azores endemic species, *H. maderensis* and *H. azorica*, follow with 20 and six samples, respectively (Figure 2).

Whereas in the Azores, all samples were of *H. azorica*, in Madeira, *H. maderensis* represented 54% of the samples, with the remaining being *H. helix* (32%), *H. algeriensis* (11%), and *H. maroccana* (3%). In mainland Portugal (Figure 3), no *H. iberica* plants were recorded in the north, while this species was the most frequent (50% of occurrences) in the south, and only sporadic (8%) in the centre. On the contrary, *H. hibernica* was much more frequent in the north and centre (44 and 54%, respectively) than in the south (5%). A parallel north–south gradient was recorded for *H. algeriensis* and *H. maroccana*; the former being more frequent in the north (40%) and less frequent in the centre (16%) and in the south (12%), and the latter showing an opposite distribution, being more frequent in the south (17%) than in the centre (6%) and in the north (4%). *Hedera helix* occurs in all mainland regions at similar frequencies (12% in the north, 16% in the centre, and 14% in the south). *Hedera canariensis* was seldom found, and only in the south.

No variegated plants were recorded among the samples belonging to *H. iberica*, *H. azorica*, *H. maderensis*, or *H. canariensis*. On the contrary, variegated plants represented 1.5% of the *H. maroccana* samples, 2.8% of *H. hibernica*, 12% of *H. helix*, and 35% of *H. algeriensis*.

Colonisation habits (high versus low density and vertical versus horizontal growth) were similar regardless of the *Hedera* species found or the region. However, these colonisation habits varied according to the type of place where plants were cultivated: whereas plants in backyards and in public and private gardens were more commonly found as dominant plants (64–68%), the frequency of samples where ivies were dominant was lower in public parks and on roadside flower beds (50–53%), and even lower in plants occurring in the edge of industrial plots (33%) (Table 1); plants on roadside flower beds and in public gardens and parks were more frequently growing horizontally only (19–31%) compared to the global average (14%), whereas plants in private gardens and backyards were rarely found growing horizontally only (0–6%). Backyards presented a higher-than-average frequency of plants growing simultaneously horizontally and vertically, whereas private gardens showed a higher-than-average frequency of plants growing vertically only (Table 2).

The frequency of each species varied according to the type of place of cultivation (Figure 4). For instance, *H. iberica* was less frequent in public gardens compared to the other species (19% versus 29–41%) but more frequent in private gardens (along with *H. maroccana*) and in public parks. Both *H. hibernica* and *H. algeriensis* were more common in backyards than the other species.

Samples were collected in backyards at a higher-than-average frequency in the north and centre of mainland Portugal (35–40%), whereas those collected in the south were more frequent in private gardens (43%). The frequency of samples collected in public gardens was even across mainland Portugal (27–28%), but that of public parks varied following no solid trend (Table 3).

A closer look at particular cases follows, in order to depict specific situations. For instance, in the Azores, all plants collected in public gardens belong to *H. azorica*, the local endemic ivy, mimicking the situation found in nature in this archipelago. In Madeira, however, a large number of samples were collected, and the situation is quite different from the Azores (Table 4). Only about half of the samples collected in Madeira belong to the endemic *H. maderensis*. Whereas *H. maderensis* was the sole species collected in public parks, *H. helix* was common in public gardens and (along with *H. algeriensis*) in backyards.

Surveys conducted in public and private gardens in the city of Guimarães (north of Portugal) revealed *H. hibernica* and *H. algeriensis* occurring at identical frequencies (48%), whereas *H. helix* occurred sporadically (4%). Similarly, surveys conducted in public and private gardens in the city of Porto (also in the north) revealed *H. algeriensis* occurring at 50% frequency, followed by *H. hibernica* (25%), *H. helix* (12.5%), and *H. maroccana* (12.5%). The results from these two examples are in line with the higher frequency of *H. algeriensis* and *H. hibernica* in the north of the country (Figure 3). Several other examples (for instance, in Boticas, Mirandela, and Vila Real) highlight the higher frequency of either *H. algeriensis* or *H. hibernica* in public and private gardens and backyards in the north region.

In the centre of the country, surveys conducted in diverse towns in the district of Aveiro (the northernmost part of this central region) revealed *H. hibernica* as the most frequent species, with *H. algeriensis* occurring sporadically. In Coimbra, *H. algeriensis* was found in parts of the city wall that could be at least one millennium old. Further south, a survey conducted in a garden in Pedrógão Grande (district of Leiria) showed *H. hibernica* as the most frequent species (50%), followed by *H. helix* (37.5%) and by *H. algeriensis* (12.5%). Not far from there, a survey conducted in Ferreira do Zêzere (district of Santarém) revealed *H. hibernica* as the most frequent species (59%), followed by *H. algeriensis* (35%) and by *H. helix* (6%). At the south edge of the centre region, a study conducted in Torres Vedras showed *H. maroccana* as the most common species (46%), followed by *H. hibernica* (27%), *H. iberica* (18%), and *H. helix* (9%), depicting the transition to the south region.

In the northern fringes of the south region, not far from Ferreira do Zêzere, a survey conducted in Torres Novas revealed *H. hibernica* as the most frequent species (38%), followed by *H. iberica* and *H. algeriensis* (25% each), and by *H. helix* (12%). Eastwards, a small survey conducted in Portalegre, Castelo de Vide, and Marvão still showed *H. hibernica* as the most common species (72%), followed by *H. algeriensis* and *H. helix* (14% each). Scattered surveys conducted in the southwestern part of the district of Santarém and in the eastern part of the district of Lisbon (municipalities of Santarém, Almeirim, Benavente, Cartaxo, Azambuja, Alenquer, and Vila Franca de Xira) show a mixed scenario, with *H. algeriensis* as the most common species (36%), closely followed by *H. iberica* (29%) and by *H. maroccana* (21%), and then by *H. helix* and *H. hibernica* (7% each). In parallel to the situation described for Coimbra, *H. maroccana* plants were recorded in areas of the Santarém castle that could date from nearly 1000 years ago. Surveys conducted in the municipalities of Almada, Seixal, Sesimbra, Setúbal, Palmela, Moita, and Alcochete (district of Setúbal) revealed *H. iberica* as the most frequent species (46%), followed by *H. maroccana* (32%) and then by *H. helix* (14%), *H. algeriensis*, and *H. hibernica* (4% each). Similarly to Santarém, *H. maroccana* plants were found at points of the Almada city walls that could date from nearly 1000 years ago. In fact, and now progressing further south, similar situations were found in Beja, Moura and Loulé, with *H. maroccana* plants occurring in association with ancient parts of these historical cities. A similar situation was found in Évora and Vila Viçosa, but in these cases, the ivy species found was *H. iberica*. In general, surveys conducted in the Alentejo and Algarve (municipalities of Elvas, Vila Viçosa, Reguengos de Monsaraz, Évora, Montemor-o-Novo, Santiago do Cacém, Ferreira do Alentejo, Odemira, Beja, Moura, Olhão, São Brás de Alportel, Loulé, and Albufeira) show both *H. maroccana* and *H. iberica* as the most frequent (30% each), followed by *H. helix* (25%), by *H. canariensis* (12%), and by *H. algeriensis* (3%).

The last three paragraphs illustrate, in further detail, the broad picture depicted in Figure 3, with the native *H. hibernica* and the exotic *H. algeriensis* being more common in the north and centre, and the native *H. iberica* and the exotic *H. maroccana* being more common in the south. The distribution of the two native species planted in gardens in urban areas mimics the distribution of these plants in nature, suggesting the use of local plant material for planting both in public and private situations.

An important fraction of the samples collected in this study represent the municipality of Lisbon (116 samples) and those in the immediate proximity: Oeiras (48 samples), Cascais and Sintra (46 samples each), Odivelas (eight), and Loures (three). In an extensive survey conducted in Carcavelos (Cascais) and Oeiras, in an area with detached houses with private gardens (private gardens represent 57% of the samples, followed by public gardens—30%), *H. iberica* was the most common species (46%), followed by *H. maroccana* (33%), *H. algeriensis* (20%), and *H. helix* (1%) (Table 5). However, these proportions vary according to the place of cultivation. Whereas *H. algeriensis* (mostly variegated types) was the most common species in public gardens in this area (48% of ivy samples in public gardens), this species was seldom used in private gardens (5% of the private garden samples), where *H. iberica* and *H. maroccana* were the most frequent (47 and 49%, respectively).

In a survey conducted in public parks in Sintra (Liberdade, Regaleira, and Seteais parks), 24 out of 26 samples were revealed to be *H. iberica*, the remaining two being *H. algeriensis*. Sintra is one of the northernmost points of the natural distribution of *H. iberica*, but, at the same time, one of the areas where this species is most abundant, and its cultivation in public parks seems to spring from its widespread occurrence in nature. Still, in the Sintra municipality, two surveys were conducted on private estates, both located to the north of the municipality, very close (ca. 500 m) to the sea front, in an area of maritime pine (*Pinus pinaster*) forest. The Assafora estate is a relatively large, single-owner property with several ornamental plants (including ivies) cultivated throughout the estate. The Colares estate is a private bungalow site, with multiple owners growing ornamental plants (ivies included) in the soil and in pots around each bungalow. In the Assafora estate, among five ivy samples collected, three species were recorded: *H. iberica*; *H. algeriensis*; *H. helix*. In the Colares estate, among six samples collected, five species were recorded: *H. iberica*; *H. hibernica*; *H. algeriensis*; *H. helix*; *H. canariensis*. Both examples demonstrate high levels of diversity, denoting informal curiosity and a craving for botanical diversity by people. The Colares estate example encompasses nearly the entire diversity of *Hedera* species recorded in continental Portugal in this study, missing out only *H. maroccana*.

In the urban area of the city of Lisbon, 113 samples were collected, 50 corresponding to private gardens, 30 to public gardens, 27 to public parks, and three each to backyards and roadside flowerbeds. The most common species was *H. iberica* (56% of samples), followed by *H. helix* (19%), *H. algeriensis* and *H. maroccana* (10% each), *H. hibernica* (3%), and *H. canariensis* (2%). The distribution of cultivated *Hedera* species in the city of Lisbon follows the trend of southern Portugal concerning the high frequency of *H. iberica* and the low frequency of *H. hibernica*, but it differs due to the fact that the frequency of *H. helix* is higher than anywhere else in the country and due to the even numbers of occurrence of the two North African species, *H. algeriensis* and *H. maroccana*. In Lisbon, *H. iberica* was the most common species in all types of cultivation places (Table 6), but the second most common species was *H. maroccana* in public gardens and backyards, and *H. helix* in private gardens and public parks. Similarly to the high species diversity recorded in the Assafora and Colares estates in Sintra, some alike situations are recorded in Lisbon: in a private garden in the Lapa neighbourhood, four distinct species (*H. iberica*, *H. helix*, *H. maroccana*, and *H. canariensis*) were recorded among five samples; both *H. helix* and *H. iberica* were recorded in the garden of a decades-long abandoned house in Alfama; both *H. iberica* and *H. maroccana* were found at the Príncipe Real garden; *H. iberica*, *H. helix*, and *H. hibernica* were recorded at the Fundação Calouste Gulbenkian garden; both *H. iberica* and *H. helix* were recorded at the Lisbon Zoo; both *H. iberica* and *H. algeriensis* were found at Quinta do Lambert. It should be noted that *H. iberica* is commonly found in areas between buildings or backyards, namely, in more declivous parts of the city. This species is native to this part of the world, and it could be that those plants were already there before the city existed. In fact, *H. iberica* is the species of ivy found in any of the three Lisbon botanical gardens (although the Ajuda Botanical Garden now hosts a collection representing nearly all known *Hedera* species worldwide).

## 3. Discussion

This study has focused on the diversity of ivy species grown in Portuguese gardens, with emphasis on geography and the type of garden. Ivies were more often found as dominating plants and growing vertically, reflecting their vigour and good adaptability to Portuguese conditions on the one hand, and on the other hand, their suitability to occupy a relatively unexplored garden niche, which is the climbers’ niche.

All samples collected in the Azores belong to the local endemism, *H. azorica*. Cultivation in gardens of the local species thus seems to be the rule in the Azores, which can be considered a valuable way to preserve the endemic *H. azorica*. The introduction of exotic species into the archipelago flora is a threat to local biodiversity, as it is estimated that about 70% of total plant taxa in this archipelago are non-native [13].

In Madeira, an opposite situation to that of the Azores was recorded, as only half of the ivy samples collected there belong to the endemic *H. maderensis*. Here, exotic species such as *H. helix*, *H. algeriensis*, and *H. maroccana* are grown in suburban/semi-rural areas, in close proximity to nature, leading to a possibility of their introduction into nature, eventually posing a risk for the survival of *H. maderensis*. In contrast to the Azores, more than half of the flora of Madeira is native [14], but in the case of *Hedera*, there is a case for concern regarding the potential dissemination of exotic ivies into nature, aggravated by the difficulty in differentiating *Hedera* species visually only. Similar situations may be occurring in other islands harbouring endemic taxa of *Hedera*. For instance, in Cyprus, home of *H. pastuchovii* subsp. *cypria*, diverse forms of *H. helix* are thought to occur [15]. In the Canary islands, the endemic *H. canariensis* may co-occur with other taxa, as a study on the misidentification of *Hedera* species [16] pointed out the incorrect use of the name “*H. canariensis*” in 40 out of 80 cases analysed (*H. canariensis* is one of the oldest *taxon* names in *Hedera*, and it has been used also to designate ivies from other parts of the Macaronesia and Western Mediterranean region).

In the present study, ivy plants grown in gardens in mainland Portugal follow a clear north–south gradient of species distribution. Among the native species, *H. hibernica* is more common in the north and centre, whereas *H. iberica* is more common in the south. Among the exotic species, *H. helix* is found at similar rates throughout the country, whereas *H. algeriensis* is more common in the north and centre, and *H. maroccana* is more common in the south. In fact, in Flora Iberica [10], the exotic *H. algeriensis* and *H. maroccana* are recognised as commonly cultivated on the Iberian Peninsula, while *H. helix* is treated in the present work as exotic, since its native distribution range does not seem to include Portugal (e.g., [17]).

*Hedera algeriensis* is the *taxon* where variegated forms were most frequently found. Although variegation may occur spontaneously, it can be taken as an indication of the use of domesticated/bred forms of *Hedera*. In this sense, no variegated forms were recorded for *H. iberica*, *H. azorica*, or *H. maderensis*, and in fact, no commercial cultivars are available for any of these taxa, so it can be assumed that the plants from these species grown in gardens were obtained from nature, either directly or through multiple rounds of cultivation. The remaining taxa found in this study, however, are commercially available, ranging from a few cultivars available (as is the case of *H. algeriensis* and *H. maroccana*) to several (*H. hibernica*) and to numerous (*H. helix*) [3]. For instance, 12% of the *H. helix* plants found are variegated forms, indicating that these plants were obtained from commerce. The fact that *H. helix* is not native to Portugal, along with the even frequency of its occurrence throughout the country, suggests that the *H. helix* plants found in this study were obtained from commercial sources. Similarly, the few samples of *H. canariensis* found were all collected in the south, possibly originating from the Spanish market. The high frequency of variegated forms in *H. algeriensis*, along with its exotic origin, indicates that these plants may also be of commercial origin. However, the species is known to have been cultivated for a long time in the Iberian Peninsula [10], which is corroborated in this study by the detection of non-variegated forms associated with ancient parts of cities of the north and centre of Portugal. A similar scenario was recorded for *H. maroccana* (a few variegated forms suggesting commercial origin, along with non-variegated forms) but, in this case, with non-variegated forms associated with ancient parts of cities in the south of the country. Finally, the most uncertain situation is that of *H. hibernica*. Cultivars of this species are available commercially, and in fact, some variegated forms were identified in this study, but the species is also native to the north and centre of the country (and also to the northern fringe on Iberia [18]), coinciding with the areas where this species was most frequently found in this study, suggesting that at least a fraction of the *H. hibernica* cultivated plants were obtained from nature. To summarise, whereas *H. iberica* plants can be assumed to be cultivated from nature and *H. helix* plants are obtained from commercial sources, some *H. algeriensis* and *H. maroccana* are from commercial sources, but some non-variegated types associated with historical cities could represent archaeophytes of North African provenance, and *H. hibernica* plants are either from commercial origin or from nature.

To a great extent, the cultivation of ivies in parks and gardens in Portugal follows the distribution of native species (*H. iberica* and *H. hibernica*), and the cultivation of *H. algeriensis* and *H. maroccana* could represent an ancient introduction of these taxa. The cultivation of *H. helix* and of variegated forms of several species (namely, *H. hibernica*, *H. algeriensis*, and *H. maroccana*) denotes influence of commercial origin, although only to a limited extent. Gardens and parks in Portugal, thus, tend to preserve and disseminate the local ivy species, adding to this some more botanical diversity. As opposed to the situation discussed for Madeira Island, the present results indicate a low risk of introduction of exotic species into Mainland Portugal. *Hedera iberica* and *H. hibernica*, the most common species in this study, are native to the areas where they were mostly found; *H. algeriensis* and *H. maroccana* are treated here as archaeophytes; *H. canariensis* was found at low frequencies; and only *H. helix* could be seen as presenting some risk of introduction. Nevertheless, the native range of this species spans nearly the whole of Europe, spanning as far west as Spain, and therefore its spread to Portugal would not represent a major shift in its distribution.

This study highlights several specific situations, either public or private, where two, three, or even more different species of *Hedera* are cultivated alongside in the same garden. Although in some cases these multiple species have diverse ornamental value (variegated versus non-variegated forms, or different leaf shapes), in several cases, there is no apparent distinction between the different species cultivated, reflecting that people in charge of such gardens are keen on gathering and preserving diversity per se. In Portugal, it is socially acceptable for someone to take a propagule of a plant from a garden for planting elsewhere (unless there is explicit information stating otherwise), provided that the source plant does not become damaged or depreciated, and ivies are particularly prone to this type of dissemination. The role of homegardens in promoting biodiversity and food security has been highlighted [19,20], as they encompass a complex relationship between food and medicine production, ecosystem services, and ornamental and cultural value [21,22]. The contribution of homegardens towards the conservation of intra-specific diversity has been addressed regarding food crops [23], but the present study illustrates a country-wide tendency to preserve diversity of an ornamental plant (in this case, at the infra-generic level) with no apparent specific value other than the diversity itself.

## 4. Materials and Methods

Ivies (*Hedera* spp.) were surveyed in parks and gardens throughout Portugal (including Azores and Madeira archipelagos). Each sample was georeferenced, and vegetative branches (with at least 10 leaves) were analysed for species identification using the taxonomic key provided by [16]. The type of garden was recorded, ranging from formal gardens to parks with less human intervention, as well as private gardens, backyards, as well as flowerbeds. Cultivation traits were also recorded (whenever possible), namely, regarding how ivies are managed (as climbers, ground covers, or allowed to grow freely). The map was created using MapChart (https://www.mapchart.net/; accessed on 12 June 2025).

## Figures and Tables

**Figure 1 plants-14-02486-f001:**
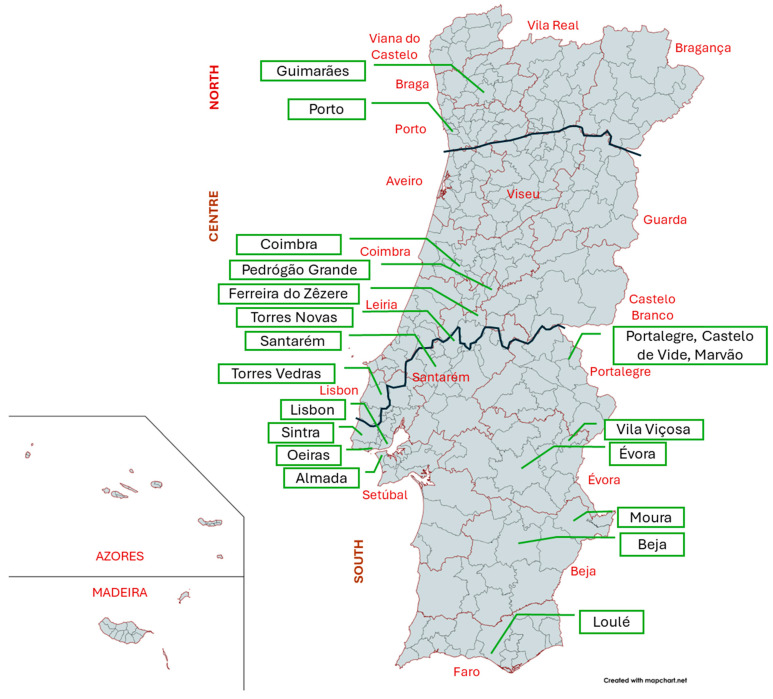
Map of Portuguese municipalities depicting the five regions under analysis (North, Centre, South, Azores, and Madeira), the 18 districts of continental Portugal (delimitated and labelled in red) and the location of municipalities pointed out in the text.

**Figure 2 plants-14-02486-f002:**
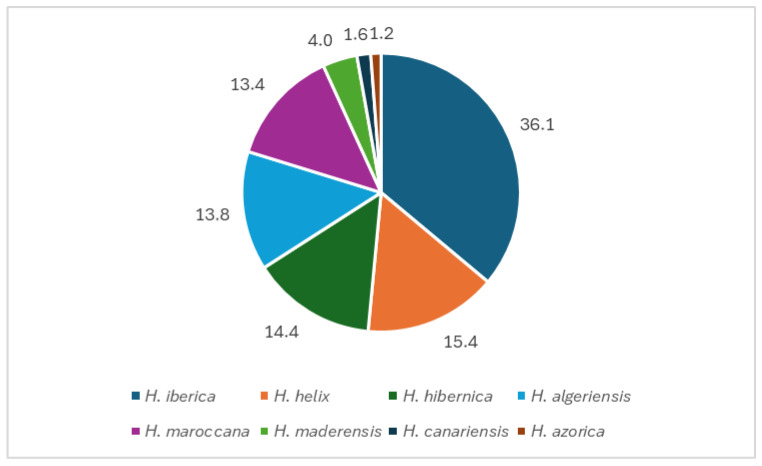
Percentage of occurrence of *Hedera* species in gardens and parks in Portugal.

**Figure 3 plants-14-02486-f003:**
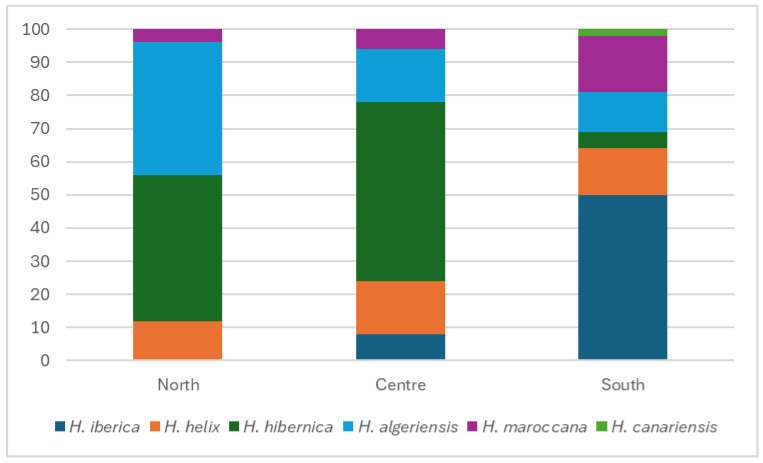
Percentage of occurrence of *Hedera* species in gardens and parks in mainland Portugal per region.

**Figure 4 plants-14-02486-f004:**
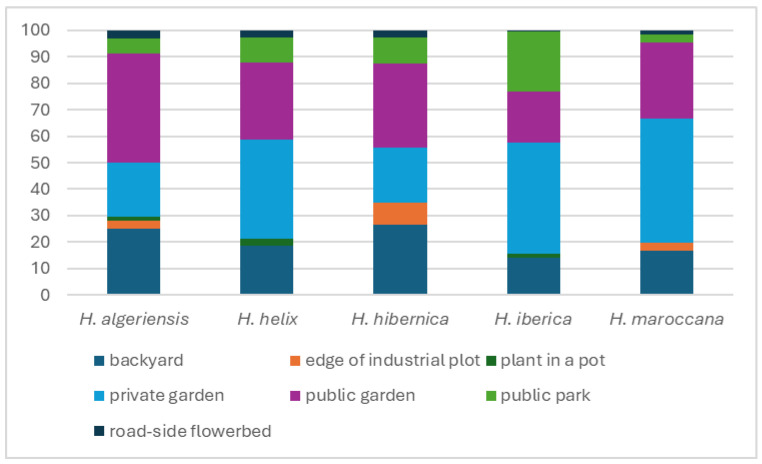
Percentage of occurrence of *Hedera* species in gardens and parks according to the type of place of cultivation (the less frequent species in this study, *H. azorica*, *H. maderensis*, and *H. canariensis*, are not included in this figure).

**Table 1 plants-14-02486-t001:** Frequency of cultivated ivy samples in Portugal according to the type of place of cultivation and to the dominance of ivies in relation to other plants; values in brackets represent percentage of each situation as a fraction of the total for each place of cultivation—table lines.

Place of Cultivation	Dominant	Intermediate	Dominated
backyard	32 (66.7)	14 (29.2)	2 (4.2)
edge of industrial plot	2 (33.3)	4 (66.7)	0 (0.0)
plant in a pot	4 (80.0)	0 (0.0)	1 (20.0)
private garden	71 (68.3)	26 (25.0)	7 (6.7)
public garden	45 (64.3)	16 (22.9)	9 (12.9)
public park	26 (53.1)	16 (32.7)	7 (14.3)
roadside flowerbed	4 (50.0)	3 (37.5)	1 (12.5)

**Table 2 plants-14-02486-t002:** Number (and frequency) of cultivated ivy samples in Portugal according to the type of place of cultivation and the growth habit (horizontal, vertical, or both) of plants. Values in brackets represent the percentage of each situation as a fraction of the total for each place of cultivation—table lines.

Place of Cultivation	Vertical	Horizontal	Both
backyard	33 (62.3)	3 (5.7)	17 (32.1)
edge of industrial plot	5 (71.4)	2 (28.6)	0 (0.0)
plant in a pot	1 (20.0)	4 (80.0)	0 (0.0)
private garden	77 (74.0)	0 (0.0)	27 (26.0)
public garden	46 (61.3)	23 (30.7)	6 (8.0)
public park	30 (62.5)	9 (18.8)	9 (18.8)
roadside flowerbed	5 (62.5)	2 (25.0)	1 (12.5)

**Table 3 plants-14-02486-t003:** Number (and frequency) of cultivated ivy samples in Portugal according to the type of place of cultivation and the geographic region. Values in brackets represent the percentage of each situation as a fraction of the total for each region—table columns.

Place of Cultivation	Azores	Madeira	North	Centre	South
backyard	0	9 (25.0)	10 (40.0)	30 (34.9)	39 (11.7)
edge of industrial plot	0	0	1 (4.0)	7 (8.1)	2 (0.6)
plant in a pot	0	0	0	1 (1.2)	5 (1.5)
private garden	0	4 (11.1)	2 (8.0)	19 (22.1)	142 (42.5)
public garden	6 (100)	16 (44.4)	7 (28.0)	23 (26.7)	90 (26.9)
public park	0	3 (8.3)	5 (20.0)	3 (3.5)	52 (15.6)
roadside flowerbed	0	4 (11.1)	0	3 (3.5)	4 (1.2)

**Table 4 plants-14-02486-t004:** Number (and frequency) of cultivated ivy samples in Madeira according to the type of place of cultivation and the species of *Hedera*. Values in brackets represent the percentage of each situation as a fraction of the total for each place of cultivation—table lines.

Place of Cultivation	*H. algeriensis*	*H. helix*	*H. maderensis*	*H. maroccana*
backyard	3 (33.3)	3 (33.3)	3 (33.3)	0
private garden	0	1 (25.0)	3 (75.0)	0
public garden	1 (6.25)	6 (37.5)	8 (50.0)	1 (6.25)
public park	0	0	3 (100)	0
roadside flowerbed	0	1 (25.0)	3 (75.0)	0

**Table 5 plants-14-02486-t005:** Number (and frequency) of cultivated ivy samples in Carcavelos (Cascais) and Oeiras area according to the type of place of cultivation and to the species of *Hedera*. Values in brackets represent the percentage of each situation as a fraction of the total for each species—table columns.

Place of Cultivation	*H. algeriensis*	*H. helix*	*H. iberica*	*H. maroccana*
plant in a pot	0	1 (100.0)	3 (8.6)	0
backyard	2 (13.3)	0	4 (11.4)	0
private garden	2 (13.3)	0	20 (57.1)	21 (84.0)
public garden	11 (73.3)	0	8 (22.9)	4 (16.0)

**Table 6 plants-14-02486-t006:** Number (and frequency) of cultivated ivy samples in Lisbon according to the type of place of cultivation and to the species of *Hedera*. Values in brackets represent the percentage of each situation as a fraction of the total for each species—table columns.

Place of Cultivation	*H. algeriensis*	*H. canariensis*	*H. helix*	*H. hibernica*	*H. iberica*	*H. maroccana*
backyard	0	0	0	0	2 (3.1)	1 (10.0)
private garden	5 (50.0)	2 (66.7)	12 (57.1)	1 (25.0)	28 (43.8)	2 (20.0)
public garden	4 (40.0)	1 (33.3)	3 (14.3)	2 (50.0)	15 (23.4)	5 (50.0)
public park	1 (10.0)	0	6 (28.6)	0	18 (28.1)	2 (20.0)
roadside flowerbed	0	0	0	1 (25.0)	1 (1.6)	0

## Data Availability

The original contributions presented in this study are included in the article. Further inquiries can be directed to the corresponding author(s).

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
