# Peer review of "Reservoirs of Biodiversity: Gardens and Parks in Portugal Show High Diversity of Ivy Species"

_plants, 2025, doi:10.3390/plants14162486_

Round 1
Reviewer 1 Report
Comments and Suggestions for Authors
This study investigates the diversity of ivy species grown in Portuguese gardens, with emphasis on geography and the type of garden.
The study follow a clear north-south gradient of species distribution. Among the native species, H. hibernica is more common in the north and centre, whereas H. iberica is more common in the south. H. algeriensis is more common in the north and centre and H. maroccana is more common in the south. H. helix is treated in the present work as exotic, since its native distribution range does not seem to include Portugal.
I think the work can be accepted after some minor revisions. I suggest some Expression linguage form and these results could support some statements in the discussion that are not currently supported.

Author Response
Line 17: Urban parks and gardens are important in multiple ways but they are often mostly composed of exotic species.
R: corrected
Line 19: and parks across diverse Portuguese cities reveal a surprisingly high level ivy species diversity, even when there is no apparent ornamental value in growing multiple species.
R: corrected
Line 22: and exotic species (mostly H. helix, H. algeriensis, H. maroccana and H. canariensis). Often, different species are cultivated side by side in the same garden, thus depicting these gardens as hidden reservoirs of biodiversity.
R: corrected
Line 35: and resilience, being quite tolerant to drought,
R: corrected
Line 39: Ivies are native to Asia and Europe, but are alien invasive on the centre-west coast of North America. They have been cultivated since Classical Antiquity as ornamental plants, and were referred as ornamental in the Iberian Peninsula by Arabs.
R: corrected
Line 47: preferences, and low water requirements for irrigation, makes the use of ivies in practical applications of landscape architecture projects beneficial, as their successful growth requires minimal maintenance [2,3].
R: corrected
Line 50: This study aims to assess the ivy species diversity cultivated in Portuguese parks
R: corrected
Line 55: In this study, we collected and identified to species level a total of 499 ivy samples
R: corrected
Line 61: Viseu – 2. In total, this comprised 25 samples from the north of the country, 86 from the centre, and 345 from the south (Figure 1).
R: corrected
Line 70: samples was collected from roadside flower beds (11), pots (6), or at the edge of industrial plots
R: corrected
Line 73: Two-thirds of the sampled ivies
R: corrected
Line 74: horizontally only, and the remaining 20% were growing both horizontally and vertically.
R: corrected
Line 88: represented 54% of the samples, with the remaining being H. helix (32%), H. algeriensis (11%), and H. 88 maroccana (3%).
R: corrected
Line 90: no H. iberica plants were recorded in the north, while this species was the most frequent
R: corrected
Line 93: A parallel north-south gradient was recorded for H. algeriensis and H. maroccana; the former being more frequent in the north (40%)
Line 96: and in the north (4%). H. helix occurred in all mainland regions at similar frequencies (12% in the north, 16% in the centre, and 14% in the south).
R: corrected
Line 97: Hedera canariensis was seldom found only in the south.
R: corrected
Line 104: H. maderensis, or H. canariensis
R: corrected
Line 107: were similar regardless of the Hedera species found or the region.
R: corrected
Line 111: lower in public parks and on roadside flower beds (50-53%), and even lower for plants occurring at the edge of industrial plots (33%)
R: corrected
Line 112: plants on roadside flower beds
R: corrected
Line 113: horizontally only (19-31%) compared to the global average (14%),
R: corrected
Line 115: horizontally only (0-6%). Backyards presented
R: corrected
Line 125: cultivation and the growth habit (horizontal, vertical, or both) of plants. Values in brackets represent the percentage
R: corrected
Line 130: H. iberica was less frequent in public gardens compared to the
R: corrected
Line 147: and the geographic region. Values in brackets represent the percentage
R: corrected
Line 152: In Madeira, however, a large number
R: corrected
Line 154: collected in Madeira belong to the
R: corrected
Line 155: sole species collected in public parks, H. helix was common in public gardens and (along with H. algeriensis) in backyards.
R: corrected
Line 159: place of cultivation and the species of Hedera. Values in brackets represent the percentage of each situation
R: corrected
Line 164: Similarly, surveys conducted in public and private gardens in the city of Porto
R: corrected
Line 166: H. helix (12.5%), and H. maroccana (12.5%).
R: corrected
Line 168: (for instance, in Boticas, Mirandela and Vila Real)
R: corrected
Line 170: or H. hibernica in public and private gardens
R: corrected
Line 171: surveys conducted in diverse towns
R: corrected
Line 173: In Coimbra,
R: corrected
Line 174: at least one millennium old.
R: corrected
Line 175: showed H. hibernica
R: corrected
Line 177: a survey conducted in Ferreira do Zêzere
R: corrected
Line 179: a study conducted in Torres
R: corrected
Line 183: conducted in Torres Novas
R: corrected
Line 184: H. algeriensis (25% each), and by H. helix (12%).
R: corrected
Line 185: survey conducted in Portalegre, Castelo de Vide, and Marvão,
R: corrected
Line 187: surveys conducted in the southwestern part [...] and in the eastern part
R: corrected
Line 189: Alenquer, and Vila Franca de Xira)
R: corrected
Line 193: Surveys conducted in the municipalities of Almada, Seixal, Sesimbra, Setúbal, Palmela, Moita, and Alcochete
R: corrected
Line 195: H. helix (14%), H. algeriensis, and H. hibernica (4% each)
R: corrected
Line 198: found in Beja, Moura, and Loulé,
R: corrected
Line 200: An alike situation was found in Évora and Vila Viçosa
R: corrected
Line 201: surveys conducted in the Alentejo and Algarve
R: corrected
Line 222: this species was seldom used
R: corrected
Line 227: and the species of Hedera. Values in brackets represent the percentage
Line 231: samples were revealed to be H. iberica
R: corrected
Line 237: maritime pine (Pinus pinaster) forest. The Assafora estate is a relatively large,
R: corrected
Line 238: throughout the estate. The Colares estate is a private bungalow site
R: corrected
Line 242: Both examples demonstrate high levels of diversity
R: corrected
Line 248: 27 to public parks, and three each to backyards 248 and roadside flowerbeds.
R: corrected
Line 254: and due to the even numbers of occurrence of the
R: corrected
Line 258: recorded in the Assafora
Line 259: some similar situations are recorded in Lisbon:in a private garden in the Lapa neighbourhood,
R: corrected
Line 262: house in Alfama;
Line 263: found in the Príncipe Real garden
R: corrected
Line 266: in areas between buildings or backyards, namely in more declivous parts of the city.
R: corrected
Line 268: part of the world, and it could be that
R: corrected
Line 269: species of ivy found in any of the three Lisbon botanical
R: corrected
Line 274: and the species of Hedera. Values in brackets represent the percentage
R: corrected
Line 282: Portuguese conditions on the one hand, and on the other hand their suitability
R: corrected
Line 285: local species thus seems to be the rule
R: corrected
Line 289: In Madeira, an opposite situation
R: corrected
Line 292: leading to a possibility of their introduction into nature,
R: corrected
Line 294: but in the case of Hedera, there is a case for concern
Line 298: occur [7]. In the Canary islands,
R: corrected
Line 314: indication of the use of domesticated/bred forms
R: corrected
Line 322: indicating that these plants were obtained from commercial sources
R: corrected
Line 341: associated with historical cities
R: corrected
Line 343: or from nature.
R: corrected
Line 346: an ancient introduction of these taxa
R: corrected
Line 347: H. algeriensis, and H. maroccana) denotes the influence of commercial origin,
R: corrected
Line 351: where two, three, or even more different species
R: corrected
Line 356: The role of homegardens in promoting biodiversity
R: corrected
Line 359: The contribution of homegardens towards the conservation
R: corrected
Line 369: with less human intervention , as well as private gardens, backyards,
R: corrected
Line 371: how ivies are managed (as climbers, ground covers, or allowed to grow freely).
R: corrected
Reviewer 2 Report
Comments and Suggestions for Authors
Urban parks and gardens are known to play an important role in modern cities and are indispensable for the development of modern urban civilization. The author has carried out a relatively systematic study of public and private gardens and parks in various cities in Portugal. The research results have certain implications for the further optimization of urban parks and gardens. However, the work has significant shortcomings in terms of overall academic quality, with limited depth and logical coherence that could be improved.
- The abstract does not highlight key quantitative findings, such as the total sample size of 499 or regional distributional differences, nor does it clearly articulate the research value. It is recommended to add key data such as ‘H. iberica accounts for 36%’ and ‘the northern region is dominated by H. hibernica (44%)’.
- In the introduction, the key progress of the research topic is not effectively presented.
- The paper emphasizes that gardens are ‘hidden reservoirs of biodiversity', but does not clarify what contribution these artificial environments actually make to species conservation compared to natural habitats (e.g. whether they promote gene flow or prevent the extinction of endangered species).
- The study is based solely on morphological identification and does not take into account hybridization between species within the same genus.
- The proportion of samples from the Lisbon region is excessively high (288/499, 57.7%), while only 25 samples (5%) come from the northern region. Does this geographical imbalance undermine the representativeness of the conclusion that ‘the northern region is dominated by H. hibernica’?
- Confounding variables were not controlled. Factors such as the age of the garden and the intensity of management (e.g. frequency of pruning) were not recorded, although these can influence species composition (e.g. new gardens tend to plant commercial varieties). This also prevents researchers from distinguishing whether the species distribution is the result of historical introductions or contemporary human selection.
- The map in Figure 1 does not indicate the density of sampling sites (only administrative districts), so it is difficult to assess the uniformity of coverage.
- The authors did not indicate whether they obtained permission to enter private gardens or collect samples (especially in protected sites such as old city walls). It is recommended that the permission number for the ethical review be added in the ‘Materials and methods’ section or in the acknowledgements.
- The depth of the discussion needs to be increased. For example, the authors did not analyze the relationship between ‘high species coexistence’ (e.g. five species in the Corrales Estate) and the risk of biological invasion.
- The authors need to present important key conclusions that are currently missing from this paper.
Author Response
Urban parks and gardens are known to play an important role in modern cities and are indispensable for the development of modern urban civilization. The author has carried out a relatively systematic study of public and private gardens and parks in various cities in Portugal. The research results have certain implications for the further optimization of urban parks and gardens. However, the work has significant shortcomings in terms of overall academic quality, with limited depth and logical coherence that could be improved.
R: We appreciate the comments by the reviewer. Indeed, the abstract and introduction sections were insufficient. We have densified the abstract and thoroughly modified the introduction. We have also added modifications to the rest of the text, mainly in discussion, in order to address the reviewer comments. Each of the comments is responded in detail next. In broad terms, however, we would like to stress that this manuscript is submitted to a special issue on ‘Plants in urban landscapes’ and it focuses on ivies collected in parks and gardens. It is part of a broader study dealing with ivies in nature in western Iberia, to be submitted soon for publication. That study will have a strong botanical focus. In the present manuscript we would like to focus more on the description and implications of species diversity, whereas the botanical/genetic traits of the samples collected in this area will be further dissected in that other study.
The abstract does not highlight key quantitative findings, such as the total sample size of 499 or regional distributional differences, nor does it clearly articulate the research value. It is recommended to add key data such as ‘H. iberica accounts for 36%’ and ‘the northern region is dominated by H. hibernica (44%)’.
R: The abstract has been consubstantiated to present more results and more precise data.
In the introduction, the key progress of the research topic is not effectively presented.
R: Thank you. We have rewritten the Introduction section, trying to raise the main points relevant to this study.
The paper emphasizes that gardens are ‘hidden reservoirs of biodiversity', but does not clarify what contribution these artificial environments actually make to species conservation compared to natural habitats (e.g. whether they promote gene flow or prevent the extinction of endangered species). The study is based solely on morphological identification and does not take into account hybridization between species within the same genus.
R: We have combined these two comments as there is a common denominator to them. We are aware that there are hybridisation events in Hedera. In our surveys we have detected 12 samples that are putative hybrids, or at least they do not fit any currently known species. These include plants with odd ploidy levels (3x, 5x and 7x), as well as hexaploid plants with stellate trichomes. In fact we have collected only one or two samples with each of these unusual traits, so we have opted to leave them out of the current study. They will be included in a separate analysis with a stronger botanical emphasis. Nevertheless, we have added text acknowledging that the artificial cultivation of plants from different species in close proximity increases the chance of hybridisation (text beginning with “The fact that plants from multiple species are grown in close proximity…”). Concerning the first part of this comment “The paper (…) does not clarify what contribution these artificial environments actually make to species conservation compared to natural habitats”. Indeed, we are not presenting here data from nature - that will be another study – so we cannot perform a direct numeric comparison to the frequency of Hedera species natural environments. However, our discussion uses information from Flora Iberica and other sources to compare the results of our study to those in nature in western Iberia. We have added some elements in Discussion (sentences starting with “As opposed to the situation discussed for Madeira island, the presents results indicate…”) to reinforce our views concerning the point raised by the reviewer. Finally, concerning the comment on endangered species, we can add that no Hedera species native to Portugal is considered endangered currently. Nevertheless, we stress (and in fact this is probably being pointed out for the first time) that only about half of the ivies found at Madeira island are of the endemic H. maderensis, pointing out a potential risk of replacement of the local species by exotic species.
The proportion of samples from the Lisbon region is excessively high (288/499, 57.7%), while only 25 samples (5%) come from the northern region. Does this geographical imbalance undermine the representativeness of the conclusion that ‘the northern region is dominated by H. hibernica’?
R: We were cautious regarding the weight of data and the conclusions made. The dominance of H. hibernica in the north and centre is highlighted by 43 samples out of 91. We agree that the south was much more heavily surveyed, but 91 samples is not a meagre figure.
Confounding variables were not controlled. Factors such as the age of the garden and the intensity of management (e.g. frequency of pruning) were not recorded, although these can influence species composition (e.g. new gardens tend to plant commercial varieties). This also prevents researchers from distinguishing whether the species distribution is the result of historical introductions or contemporary human selection.
R: Ivies are very good at hiding their age. Unlike an old oak or olive tree, even if an ivy plant is centuries old, the way it propagates, it could look as if it was planted only 30 years ago. In some cases, we suspect that ivies were there before the garden was planted or even before there was any urban occupation. This hypothesis is provided in the manuscript, but with due cautiousness. We fully agree that data on the history and age of each garden, as well as on management, would make the study more robust. In most cases, this type of information was unprecise, mainly because many gardens had several waves of reforms and new plantings, rendering unclear when ivies were planted. Therefore, we could not make use of such variables.
The map in Figure 1 does not indicate the density of sampling sites (only administrative districts), so it is difficult to assess the uniformity of coverage.
R: Sampling was not uniform, as it was restricted to urban areas. The purpose of the map in Figure 1 is only to enable the reader to locate the geographic location of towns and cities mentioned in the text, without the need to look for them using external sources.
The authors did not indicate whether they obtained permission to enter private gardens or collect samples (especially in protected sites such as old city walls). It is recommended that the permission number for the ethical review be added in the ‘Materials and methods’ section or in the acknowledgements.
R: Surveys were conducted according to national rules and ethical standards, and as part of the professional/academic activity of the authors and of colleagues that are acknowledged for supplying samples. To this end, we have added the following sentence in Discussion “In Portugal, it is socially acceptable for someone to take a propagule of a plant from a garden for planting elsewhere (unless there is explicit information stating otherwise), provided that the source plant doesn’t become damaged or depreciated, and ivies are particularly prone for this type of dissemination.” A small note on old city walls. When we say “parts of the city wall that could be at least one millennium old” it doesn’t necessarily mean that the wall is standing there for 1000 years. It means that it is known that in that place there has been a wall since Arab times but for sure (this is according to historians) the wall has collapsed and been rebuilt many times and most likely in the 1940s, when many monumental sites have been reconstructed under specific historical and political circumstances (see, for instance, https://doi.org/10.37935/iha.aon.2022.0004). Ivies would easily survive the crumbling and rebuilding of such walls.
The depth of the discussion needs to be increased. For example, the authors did not analyze the relationship between ‘high species coexistence’ (e.g. five species in the Corrales Estate) and the risk of biological invasion.
R: The Discussion section was strengthened. Specifically, the risk of biological invasion has been given a more direct look. Please also refer to another reply regaring this.
The authors need to present important key conclusions that are currently missing from this paper.
R: To this end, we have strengthened the discussion and the abstract.
Round 2
Reviewer 2 Report
Comments and Suggestions for Authors
The author has made the necessary revisions, and the current version can be published.